# In Vitro Synergistic Inhibitory Effects of Plant Extract Combinations on Bacterial Growth of Methicillin-Resistant *Staphylococcus aureus*

**DOI:** 10.3390/ph16101491

**Published:** 2023-10-20

**Authors:** Jae-Young Jeong, In-Geun Jung, Seung-Hoon Yum, You-Jin Hwang

**Affiliations:** 1Department of Health Sciences & Technology, Gachon Advanced Institute for Health Sciences & Technology, Gachon University, Incheon 21999, Republic of Korea; jini656565@naver.com; 2Department of Biomedical Engineering, College of Health Science, Gachon University, Incheon 21936, Republic of Korea; sun00060@gachon.ac.kr (I.-G.J.); markyum@gachon.ac.kr (S.-H.Y.)

**Keywords:** synergistic effects, antibacterial, medicinal plants, antibiotic resistance, methicillin-resistant *Staphylococcus aureus*

## Abstract

Methicillin-resistant *Staphylococcus aureus* (MRSA) is one of the most common pathogens of healthcare-associated infections. Medicinal plants have long been used in the traditional treatment of diseases or syndromes worldwide. Combined use of plant extracts could improve the effectiveness of pharmacological action by obtaining synergism, acting on multiple targets simultaneously, reducing the doses of individual components, and minimizing side effects. We aimed to investigate the synergistic inhibitory effects of selected medicinal plants (*Caesalpinia sappan* L. (CS), *Glycyrrhiza uralensis* Fisch. (GU), *Sanguisorba officinalis* L. (SO), and *Uncaria gambir* Roxb. (UG)) on the bacterial growth of MRSA and its clinical isolates. SO and UG extracts generated the best synergistic interaction as adjudged by checkerboard synergy assays. MICs of the individual extracts decreased 4-fold from 250 to 62.5 μg/mL, respectively. The SO + UG combination was further evaluated for its effects on bacterial growth inhibition, minimum bactericidal/inhibitory concentration (MBC/MIC) ratio, and time-kill kinetics. The results indicate that the SO + UG combination synergistically inhibited the bacterial growth of MRSA strains with bactericidal effects. SO + UG combination also exhibited more potent effects against clinical isolates. In multistep resistance selection experiments, both standard and isolates of MRSA showed no resistance to the SO + UG combination even after repeated exposure over fourteen passages. Our data suggest that using plant extract combinations could be a potential strategy to treat MRSA infections.

## 1. Introduction

Methicillin-resistant *Staphylococcus aureus* (MRSA) is one of the most prevalent pathogens of healthcare-associated (HA) infections [1]. MRSA has become a serious threat to global health and can cause mild to invasive, life-threatening infections. It is responsible for increased morbidity and mortality, length of stay, and economic burden [1,2,3]. The overall proportion of MRSA isolates exceeded 20% in all World Health Organization (WHO) regions, and even exceeded 80% in some reports [3]. In hospital settings, the prevalence of MRSA has been reported to be 70–80% in Asian countries, more than in Europe (25%) [4]. In particular, the proportion of MRSA in HA isolates was 73.3% in South Korea [5].

MRSA is a major threat among antibiotic-resistant agents, causing ~19,000 deaths with a healthcare cost of USD 3–4 billion annually in the US. [6]. The ability of MRSA to tolerate conventional antibiotics leads to difficult-to-treat infections and limits the therapeutic options available [4,7]. Methicillin resistance in *Staphylococci* is mediated by the *mec*A gene, which encodes a modified penicillin-binding protein 2a (PBP 2a) that results in resistance to beta-lactam antibiotics by causing a low binding affinity [8,9]. Furthermore, over time, MRSA has developed resistance to other antibiotic classes, including fluoroquinolone, macrolide, aminoglycoside, and clindamycin [10]. With the growing problem of antibiotic resistance, novel antibiotic agents with different mechanisms of action are urgently needed to control MRSA infections.

Many plant species found to possess medicinal values have long been used in the traditional treatment of diseases or syndromes worldwide [11,12]. Medicinal plants are still being provided as traditional medicines to 70–95% of the population in developing countries. They are also utilized, either directly or indirectly, in at least 25% of all modern medicines [13]. Medicinal plants have various bioactive compounds, such as alkaloids, flavonoids, phenolic compounds, steroids, tannins, terpenoids, and other secondary metabolites, which act remarkably on parasites and pathogens [12,14]. Plant-derived compounds possess unique pharmacological properties such as low cost, less toxicity, fewer side effects, and less likely to develop resistance [15,16,17].

Synergism is the interaction of two or more drugs that produces a greater influence than either individually [18]. Synergism is preferred to treat infections associated with multidrug-resistance (MDR) or those at risk of treatment failure with a single drug because plant extracts in combination provide more benefits than what is generally available alone [19,20]. Combined use of plant extracts could improve the effectiveness of pharmacological action by obtaining synergism, acting on multiple targets simultaneously, reducing the doses of individual extracts, and minimizing side effects [21,22]. However, it is important to know which of the four possible effects (synergism, partial synergism, addition, or antagonism) of a therapeutic or even toxic response leads to the combined effect of plant extracts and to optimize the appropriate proportion that produces a more effective therapeutic effect [23,24].

A previous study by our team has reported the inhibitory effects of medicinal plants on the bacterial growth of MRSA. We also selected four medicinal plants (*Caesalpinia sappan* L. (CS), *Glycyrrhiza uralensis* Fisch. (GU), *Sanguisorba officinalis* L. (SO), and *Uncaria gambir* Roxb. (UG)) based on their potent effects in that study [25]. The present study aimed to investigate the synergistic effects of selected medicinal plants against MRSA strains, including clinical isolates. To the best of our knowledge, this is the first report to investigate the synergistic inhibitory effects of selected medicinal plants on the bacterial growth of MRSA strains.

## 2. Results and Discussion

In this study, we confirmed the synergistic inhibitory effects of selected medicinal plants on the bacterial growth of MRSA and its clinical isolates. The selection of medicinal plants was established based on minimum inhibitory concentrations (MICs) for MRSA strains reported in our previous study [25]. MIC values of medicinal plants were determined using the broth microdilution method as follows: CS (62.5 μg/mL), GU (250 μg/mL), SO (250 μg/mL), and UG (250 μg/mL) [25]. Information on the plants and their pharmacological uses is presented in Table 1 [26,27,28,29,30].

Based on the UPLC analysis, the major compounds of selected medicinal plants were tentatively identified. Representative chromatograms obtained from the UPLC analysis are shown in Figure 1. The chromatograms recorded at different detection wavelengths are presented in Appendix A. For each peak, we tentatively identified five compounds as follows: Peak 1 (Brazilin, observed RT: 2.98 min; formula: C_16_H_14_O_5_; molecular weight: 286.28 g/mol), Peak 2 (Protosappanin B, observed RT: 3.10 min; formula: C_16_H_16_O_6_; molecular weight: 304.30 g/mol), Peak 3 (Liquiritin apioside, observed RT: 4.35 min; formula: C_26_H_30_O_13_; molecular weight: 550.51 g/mol), Peak 4 (Glycyrrhizin, observed RT: 9.48 min; formula: C_42_H_62_O_16_; molecular weight: 822.94 g/mol), and Peak 5 (Catechin, observed RT: 3.07 min; formula: C_15_H_14_O_6_; molecular weight: 290.27 g/mol). Table 2 summarizes detailed information on each compound identified or deduced based on data reported in the literature [31,32,33,34,35,36]. The chemical structures of these compounds are shown in Figure 2.

The antimicrobial activities of medicinal plants are attributed to their ability to produce several secondary metabolites with antimicrobial properties [37]. These activities are dependent not only on the presence of secondary metabolites but also on their concentration and the possible interactions with other components [37,38]. Therefore, it is important to identify compounds involved in a specific pharmacological action and investigate their interactions with other compounds. Brazilin (Peak 1) and protosappanin B (Peak 2) have been reported as the major compounds in CS [32]. Protosappanin displayed antibacterial activities with MIC at 128 µg/mL against both *S. aureus* and MRSA [39]. Especially, brazilin showed remarkable activities against antibiotic-resistant bacteria, including MRSA, vancomycin-resistant enterococci, and multidrug resistant *Burkholderia cepacia*, with MIC values ranging from 4 to 32 µg/mL [38]. Liquiritin apioside (Peak 3) and glycyrrhizin (Peak 4) are the main bioactive components with major pharmacological activities in GU [40]. Glycyrrhizin inhibited the growth of clinical isolates of MRSA and MSSA, with MIC ranging from 32 to 512 μg/mL and 16 to 512 μg/mL, respectively [41]. Catechin (Peak 5) has been reported as the major compound in UG [42]. MIC values of catechin ranged from 78.1 to 156.2 μg/mL against clinical isolates of MRSA [43]. Flavonoids such as brazilin, protosappanin B, liquiritin apioside, and catechin can have various antibacterial mechanisms against bacteria. They inhibit biofilm formation, cell envelope synthesis, nucleic acid synthesis, and ATP synthesis and damage the bacterial respiratory chain, membrane bilayer, and membrane proteins [44]. As a terpenoid, glycyrrhizin’s antibacterial activity is due to the disruption of membranes, anti-quorum sensing, and inhibition of protein and ATP synthesis [45]. However, major active compounds involved in the antibacterial activity of SO extract were not identified [46]. This may be because the antibacterial effect of SO extract is not caused by a specific compound but rather by interactions between several compounds in the extract. Therefore, further studies are needed to identify unknown antibacterial compounds of SO extract and elucidate their antibacterial mechanism.

Antibacterial bioassays were conducted to evaluate the inhibitory effects of plant extracts alone and in combination. Two reference strains (MSSA and MRSA) and two clinical isolates (MDRSA and MRSA) were employed in these assays. Methicillin-susceptible *S. aureus* (MSSA; *S. aureus* ATCC 29,213) showed no resistance to 10 different antibiotic discs used in the antibiotic susceptibility testing: ampicillin (Amp; 10 µg), methicillin (Meth; 5 µg), penicillin G (Pen; 10 IU), kanamycin (Kan; 30 µg), gentamicin (Gen; 10 µg), streptomycin (Strep; 10 µg), tetracycline (Tet; 30 µg), erythromycin (Eryth; 15 µg), vancomycin (Van; 30 µg), and chloramphenicol (Chl; 30 µg) (Liofilchem, Teramo, Italy). The antibiotic resistance profiles of MRSA strains were as follows: MRSA (ATCC 33,591; Amp, Meth, Pen, Kan, Eryth, Strep, Tet, Gen, and Chl), MDRSA (CI-2; Amp, Meth, Pen, Kan, Eryth, Strep, Tet, and Gen), and MRSA (CI-21; Amp, Meth, Pen, Kan, Strep, and Gen) [25]. MDR was defined as strains resistant to at least one antibiotic in three or more different antibiotic classes [47]. All MRSA strains were resistant to antibiotics of beta-lactam and aminoglycoside classes. The most resistant isolate was CI-2, which was resistant to 8 out of 10 tested antibiotics and only sensitive to vancomycin and chloramphenicol. As most MDRSA isolates have been reported to be sensitive to chloramphenicol in the previous study [48], CI-2 was found to be susceptible to chloramphenicol.

MICs of individual extracts were determined by the broth microdilution method. According to Appendix A and Table 3, the ethanol extracts of selected medicinal plants showed significant inhibitory effects with MIC values ranging from 62.5 to 250 μg/mL. Among the tested extracts, CS extract had the lowest MIC (62.5 μg/mL) value against all tested strains, while other tested extracts had equal MIC (250 μg/mL) values. The inhibitory effects of selected medicinal plants were relatively more potent than most plant extracts reported in large-scale screening studies [10,11,49]. Furthermore, the susceptibility of each tested strain to different extracts showed no significant differences. These results suggest that plant extracts could be active with different antibacterial mechanisms and target sites compared to conventional antibiotics toward bacterial strains regardless of their antibiotic resistance patterns [50].

Combined use of plant extracts can cause different interactions of natural compounds because each extract contains diverse types of compounds. The enhanced antibacterial activity of plant extract combinations is well-known, as has been reported in previous literature [12,23,51]. However, some interactions decrease the efficacy of plant extract combinations by neutralizing each other, forming inactive complexes, and/or acting competitively for the same molecular target [52,53,54]. Therefore, it is necessary to confirm the influence of the combination of plant extracts. Checkerboard synergy assays were performed to evaluate the synergistic effects of selected medicinal plants (Appendix A). Fractional inhibitory concentrations (FICs) and their interpretations are presented in Table 4. SO and UG extracts generated the best synergistic interactions (FICI = 0.5). Their MICs decreased 4-fold, respectively. On the other hand, the GU + SO combination showed additive effects (FICI = 1). Other tested combinations showed partial synergistic effects (FICI = 0.625). No antagonism was found in any plant extract combinations. Among individual extracts used in combination, the CS extract showed the highest MIC reduction with an 8-fold decrease (MIC = 7.81 μg/mL), but there was no highest synergistic effect (Table 4). These results suggest that the potent activity of one extract might not necessarily lead to a high synergy with another extract. Antibacterial mechanisms of plant extract combinations are not fully understood yet. Thus, we conducted further antibacterial analysis to elucidate the inhibitory effects of the SO + UG combination, showing the best synergism.

To assess the efficacy of the SO + UG combination in inhibiting MRSA strains, the results of checkerboard analysis are presented as heatmaps indicating the percentage of bacterial growth inhibition based on optical density at 595 nm (OD_595_) values (Figure 3a). Darker regions represent higher bacterial cell density. FICIs of SO and UG extracts were 0.5 for all tested strains (Table 4). These results indicate that their synergistic effects existed regardless of the different resistance patterns of the tested strains. According to cell viability assays performed in previous studies, SO and UG extracts showed no cytotoxicity and were safe [55,56]. However, further studies are needed on the influences of the combined effects of SO and UG extracts on toxicity.

Bacterial growth curves have been used to investigate the growth and death of bacteria over a wide range of antibacterial concentrations and to assess the effects of antibacterial agents over time [57]. We monitored the inhibitory effects of the SO + UG combination on the bacterial growth of MRSA strains. The results are presented in Figure 3b. The SO + UG combination affected bacterial growth in a time- and concentration-dependent manner. At the above MIC, the SO + UG combination completely inhibited bacterial growth. The SO + UG combination showed stronger inhibitory effects than the individual extracts of sub-MIC (125 μg/mL), even at 1/2 MIC. At 1/2 MIC, the time lag with the SO + UG combination reaching the exponential phase was changed from 8 h to 12 h against clinical isolates (CI-2 and CI-21) compared to the control group. For standard strains (ATCC 29,213 and ATCC 33,591), the reaching time to the exponential phase was remarkably delayed from 8 h to 14 h in the presence of the SO + UG combination.

Bactericidal activity is of clinical importance because bacterial killing is predicted to produce a faster resolution of infection and improved clinical outcomes. More rapid elimination of bacterial pathogens should also minimize the possible emergence of resistance and spread of infections [58]. The MBC/MIC ratio determines whether a drug is bactericidal or bacteriostatic. If the MBC/MIC ratio is ≤4, the effect is considered bactericidal, but if the MBC/MIC ratio is >4, the effect is defined as bacteriostatic [59]. The MBC/MIC ratio is shown in Figure 3c. The SO + UG combination was considered bactericidal against all tested strains as its MBC/MIC ratio was 2. SO extracts showed bacteriostatic effects with an MBC of 2000 μg/mL and MIC of 250 μg/mL, while UG extracts were bactericidal with an MBC of 500 μg/mL and MIC of 250 μg/mL.

A time-kill kinetic analysis was conducted to determine the killing kinetics of the SO + UG combination. Time-kill curves are shown in Figure 3d. At 2 MIC, the SO + UG combination reduced the number of viable bacterial cells by more than 2 log_10_ within 6 h and completely eradicated the cells within 12 h. The combination especially showed stronger bactericidal effects on clinical isolates of MRSA than on standard strains. Although the SO + UG combination of MIC showed a slow log_10_ decline of viable bacterial cells without completely eradicating the cells even after 24 h, it showed a reduction of CFU/mL value of more than 2 log_10_. The combinations at higher concentrations caused more rapid bacterial death. The kinetics of the SO + UG combination killing the bacterial strains was time- and concentration-dependent. It is consistent with our results obtained from the bacterial growth curves. Our data demonstrate that the SO + UG combination could inhibit bacterial growth by acting as a bactericidal agent against MRSA strains.

Disc diffusion assays were performed to evaluate the antibacterial activity of SO + UG combination at a concentration of 2 mg/disc. The diameter of inhibition zones was measured and recorded by a representative photograph and comparative graph (Figure 4). SO, UG extracts alone, and their combination exhibited antibacterial activities against MRSA strains. Distilled water (DW) showed no antibacterial activity. All tested strains were highly susceptible to the SO + UG combination with the largest inhibition zones (24.47–25.53 mm). The UG extract also showed remarkable antibacterial activities with diameters of 20.20–21.07 mm for inhibition zones. In contrast, the SO extract had poor activities with diameters of 9.17–10.87 mm. The SO + UG combination inhibited bacterial growth more effectively than SO or UG extracts alone. In addition, SO and UG extracts showed an equal MIC of 250 μg/mL in our data, but significant differences were observed for the diameters of inhibition zones. This phenomenon could be due to the structural diversity of compounds present in plant extracts. The quantity, diversity, and biological properties of secondary metabolites from medicinal plants differ among the species of plants [60]. The disc diffusion method is dependent on several factors that contribute to the degree of diffusion, such as the polarity, hydrophilicity or hydrophobicity, and molecular weight (MW) of the test substances [61,62]. Thereby, we speculate that different physicochemical properties of diverse compounds in the plant extracts might cause differences in antibacterial activity by the disc diffusion method.

Antibiotics revolutionized the practice of medicine by providing a cure and decreasing the morbidity and mortality of numerous infectious diseases. However, these achievements are threatened by the emergence of antimicrobial resistance (AMR). AMR refers to the ability of microbial pathogens to avoid or delay death upon exposure to antibiotics predicted to kill them [63,64]. In multistep resistance selection experiments, we confirmed the ability of MRSA to develop resistance to the SO + UG combination and rifampicin after repeated exposures. MRSA strains were serially passaged at 24 h intervals in the presence of each sample for up to fourteen passages. The results are shown in Figure 5. The MRSA standard (ATCC 33,591) developed resistance to rifampicin after four passages with a 4-fold increase in MIC. MRSA isolates (CI-2 and CI-21) rapidly developed resistance to rifampicin after the second passage, with a 4-fold increase in MIC, respectively. After fourteen passages, the MIC of rifampicin for ATCC 33,591, CI-2, and CI-21 increased to 64, 1024, and 1024 folds, respectively. However, no resistance was observed for the SO + UG combination over the fourteen passages (MIC increased by up to 2-fold). These results indicate that MRSA strains could not easily develop resistance to the SO + UG combination. This could be due to the bactericidal activity of the SO + UG combination toward MRSA. Elimination of pathogens rather than inhibition eradicates the resistance mutations that could occur due to antibiotic pressure [65]. Thereby, plant extract combinations such as the SO + UG combination may be a promising candidate to overcome antibiotic resistance.

## 3. Materials and Methods

### 3.1. Plant Materials

Four medicinal plants (*Caesalpinia sappan* L., *Glycyrrhiza uralensis* Fisch., *Sanguisorba officinalis* L., and *Uncaria gambir* Roxb.) were selected based on data reported in our previous study [25]. They were purchased from Samhong Medicinal Herb Market (Seoul, Republic of Korea).

### 3.2. Preparation of Plant Extracts

Plant materials were blended to powder using a home grinder and extracted with 70% ethanol with shaking (110 rpm) for 24 h. The ratio of plant materials to solvent was 1:10 (*w/v*). Crude extracts were then centrifuged at 3000 rpm for 30 min. Supernatants were concentrated under reduced pressure using a rotary vacuum evaporator WEV-1001V (Daihan Scientific Co., Wonju, Republic of Korea). The concentrated residue was subsequently dissolved in 10% dimethyl sulfoxide (DMSO; Sigma Chemical Co., St. Louis, MO, USA) and filtered through Whatman filter paper No. 2 (Whatman, Kent, UK) to obtain ethanol extracts. All prepared extracts were collected into conical tubes and stored in a refrigerator at 4 °C until further use.

### 3.3. Ultra-Performance Liquid Chromatography (UPLC) Analysis

To tentatively analyze plant extracts, we performed an ultra-performance liquid chromatography (UPLC) analysis on an AQUITY UPLC I-Class system (Waters Corporation, Milford, MA, USA) using an ACQUITY UPLC BEH C18 1.7 μm column (2.1 × 100 mm). The mobile phase was composed of distilled water with 0.1% formic acid (A) and acetonitrile with 0.1% formic acid (B). The eluent was set as follows: 0 min 92% (A)/8% (B), 1.0 min 92% (A)/8% (B), 16.0 min 30% (A)/70% (B), 17.0 min 0% (A)/100% (B), 19.0 min 0% (A)/100% (B), 19.3 min 92% (A)/8% (B), and 22.0 min 92% (A)/8% (B) at a 0.4 mL/min flow rate (Appendix A). The column temperature was kept at 35 °C, the injection volume was 1 μL, and the detection wavelength was set at max plot (190–500 nm), 210 nm, 254 nm, 280 nm, and 360 nm. For MS detection, the following MS conditions were set for both positive and negative electrospray ionization (ESI) modes: desolvation gas (N_2_), flow rate 800 L/h, desolvation gas temperature 350 °C, source temperature 110 °C, capillary voltage 300 V, cone voltage 40 V, and *m*/*z* range 100–1500 Da (Appendix A). The identification of compounds was based on mass and UV-Vis spectra in comparison with previous literature for each origin plant.

### 3.4. Bacterial Culture

Standard strains of *S. aureus* (ATCC 29,213) and MRSA (ATCC 33,591) were used in the present study. These strains were purchased from the American Type Culture Collection (ATCC; Manassas, VA, USA). MRSA isolates (CI-2 and CI-21) were originally obtained from clinical specimens and identified at Gachon University Gil Medical Center (Incheon, Republic of Korea) [66]. These isolates were preserved in a −80 °C freezer in 20% glycerol (*v*/*v*) until further use. Each bacterium was initially cultivated on a brain heart infusion (BHI; Kisan Bio, Seoul, Republic of Korea) plate. A single colony was picked from each plate and pre-cultured in BHI broth at 37 °C for 24 h prior to assays. Bacterial stocks were subcultured every 3–4 weeks to maintain bacterial viability.

### 3.5. Determination of Minimum Inhibitory Concentration (MIC)

MIC was determined using the broth microdilution method described by Bostanci et al. [67] with slight modifications. Briefly, 200 μL of the sample was inoculated to the first wells of a 96-well microplate and serially diluted 2-fold. Then, 100 μL of each bacterial suspension (1 × 10^6^ CFU/mL) was added to each well. The microplate was incubated at 37 °C for 18 h and continuously monitored for bacterial growth. The OD was measured at 595 nm using a spectrophotometer (Multiskan FC; Thermo Fisher Scientific, Waltham, MA, USA). MIC was defined as the lowest concentration that inhibited the visible growth of bacteria.

### 3.6. Determination of Minimum Bactericidal Concentration (MBC)

To determine MBC, the MIC test was repeated, as mentioned above. Then, 50 µL of suspension was taken from the well that inhibited visible growth of bacteria. The suspension was transferred to a new microplate, and 150 µL of BHI broth was added to each well. The microplate was incubated at 37 °C for 18 h. MBC was defined as the lowest concentration that killed 99.9% of bacteria.

### 3.7. Checkerboard Synergy Assay

Synergistic effects of plant extract combinations were evaluated with a checkerboard synergy assay to obtain FIC values. One extract was serially diluted along the abscissa, while another extract was serially diluted along the ordinate. The total volume of the combination was 100 μL per well. Then 100 μL of each bacterial suspension (1 × 10^6^ CFU/mL) was added to each well of a 96-well microplate. The microplate was incubated at 37 °C for 18 h. The OD was measured at 595 nm with a spectrophotometer. FICI was calculated using the following equation: FICI = FIC_A_ + FIC_B_, where FIC_A_ was MIC of extract A in combination/MIC of extract A alone, and FIC_B_ was MIC of extract B in combination/MIC of extract B alone. Results were interpreted as synergistic interaction (FICI ≤ 0.5), partial synergy (0.5 < FICI ≤ 0.75), additive interaction (0.75 < FICI ≤ 1.0), indifferent (1.0 < FICI ≤ 4.0), or antagonistic interaction (FICI > 4.0) [68].

### 3.8. Time-Kill Kinetic Analysis

A time-kill kinetic analysis was performed to confirm the killing potencies of the SO + UG combination, according to Mohamed et al. [69]. Each bacterial suspension (1 × 10^6^ CFU/mL) was inoculated to BHI broth containing the SO + UG combination with different concentrations (2MIC, MIC, and 1/2MIC). The mixture was incubated in a shaker incubator at 120 rpm and 37 °C. Then 100 μL of the incubated mixture was transferred to BHI plates at 0, 6, 12, 18, and 24 h. The plates were incubated at 37 °C for 24 h. After incubation, a single colony was counted and calculated as log_10_ CFU/mL.

### 3.9. Disc Diffusion Assay

A disc diffusion assay was performed to evaluate the antibacterial activity of the SO + UG combination using the Kirby-Bauer disc diffusion method [70]. Each bacterial suspension was adjusted to the McFarland 0.5 turbidity standard and swabbed onto BHI plates. Then, 100 μL of samples were loaded onto paper discs (8 mm/diameter). These discs were gently placed onto the plates. After 24 h incubation, the diameter of the inhibition zone was measured and recorded.

### 3.10. MultiStep Resistance Selection against MRSA

To assess the potential of MRSA to develop resistance to the SO + UG combination and rifampicin after repeated exposure, we performed multistep resistance selection experiments according to Mohammad et al. [71]. The MIC test was conducted for the samples as described above. MRSA strains were exposed to BHI broth containing different samples. The strains in sub-MIC wells were repassaged at 24 h intervals for up to fourteen passages. Resistance was defined as a more than 4-fold increase in MIC compared to the initial MIC [71].

### 3.11. Statistical Analysis

Statistical analysis was conducted with Prism 5 (GraphPad Software Inc., San Diego, CA, USA) and SigmaPlot version 12.0 (Systat Software, San Jose, CA, USA). All data are presented as mean ± standard deviation (±SD) from triplicate experiments. Statistical differences were assessed by analysis of variance (ANOVA). A *p*-value of less than 0.05 was considered statistically significant.

## 4. Conclusions

We confirmed the synergistic inhibitory effects of selected medicinal plants on the bacterial growth of MRSA and its clinical isolates. All tested combinations showed enhanced inhibitory effects except for the GU + SO combination. SO and UG extracts generated the best synergism as adjudged by checkerboard synergy assays. MICs of the individual extracts decreased by 4-fold, respectively. In further antibacterial analysis, the SO + UG combination showed significant bacterial growth inhibition with bactericidal effects. The SO + UG combination also exhibited more potent effects against clinical isolates. Both standard and isolates of MRSA showed no resistance to the SO + UG combination even after repeated exposure over fourteen passages. Our data demonstrate that using plant extract combinations could be a potential alternative to conventional antibiotics for the treatment of MRSA infections. Further studies are needed to identify the mechanism of action, toxicity, and safety of plant-extract combinations. Especially, identification and characterization of unknown antibacterial compounds should be conducted to elucidate the synergistic interaction of plant extract combinations.

## Figures and Tables

**Figure 1 pharmaceuticals-16-01491-f001:**
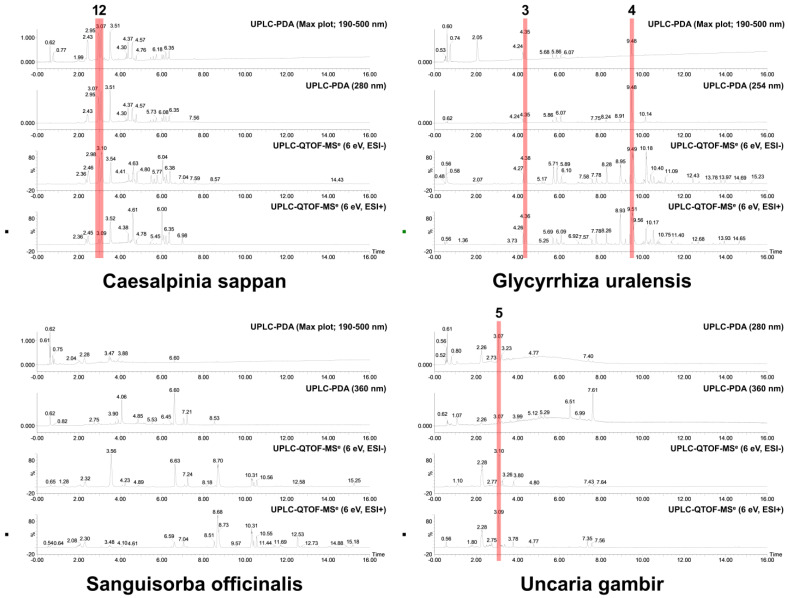
Representative UPLC chromatograms of 70% ethanol extracts (3 mg/mL) from selected medicinal plants. The tentatively identified compounds are as follows: (1) brazilin; (2) protosappanin B; (3) liquiritin apioside; (4) glycyrrhizin; (5) catechin. The analysis conditions were set as follows: column, ACQUITY UPLC BEH C18 1.7 μm column (2.1 × 100 mm); column temperature, 35 °C; flow rate, 0.4 mL/min; injection volume, 1 μL; detection wavelength, max plot (190–500 nm), 210 nm, 254 nm, 280 nm, and 360 nm. The mobile phase gradient conditions were set as described in Appendix A.

**Figure 2 pharmaceuticals-16-01491-f002:**
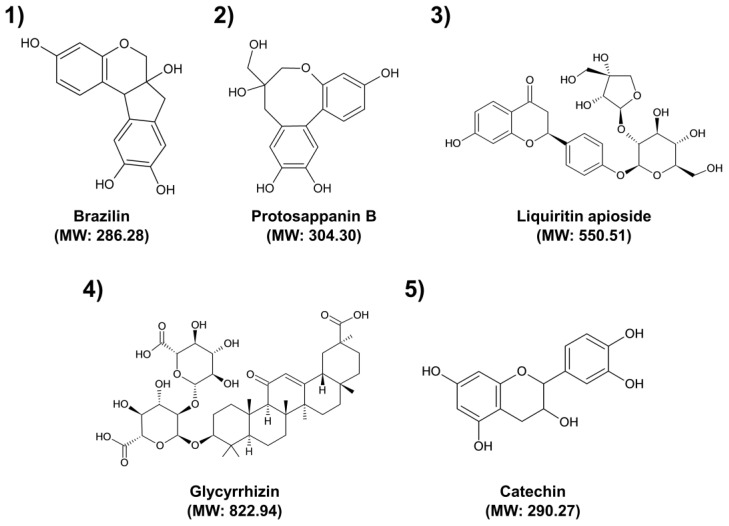
Chemical structures of the tentatively identified major compounds. (1) Brazilin; (2) Protosappanin B; (3) Liquiritin apioside; (4) Glycyrrhizin; (5) Catechin. MW: Molecular weight.

**Figure 3 pharmaceuticals-16-01491-f003:**
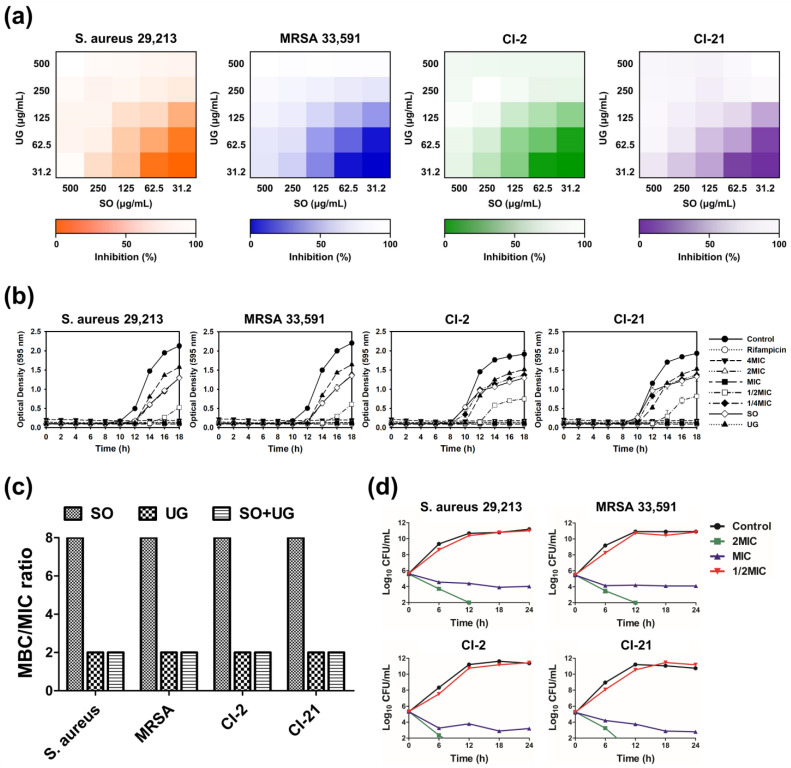
The SO + UG combination synergistically inhibits bacterial growth of MRSA isolates. (**a**) Heatmaps of checkerboard synergy assays for the SO + UG combination. The results are presented as the percentage of bacterial growth inhibition based on OD_595_ values. Darker regions represent higher bacterial cell density. (**b**) Influences of the SO + UG combination on bacterial growth of MRSA strains. The concentration of the individual extracts was 125 μg/mL (sub-MIC). (**c**) MBC/MIC ratio of SO, UG extracts alone, or their combination. (**d**) Time-kill curves of the SO + UG combination against MRSA strains. MBC/MIC ratio ≤ 4 and time-kill curves indicate bactericidal effects of the SO + UG combination following data in (**c**,**d**). All data are presented as mean ± standard deviations of experiments performed in triplicate, with *p* < 0.05 indicating statistical significance.

**Figure 4 pharmaceuticals-16-01491-f004:**
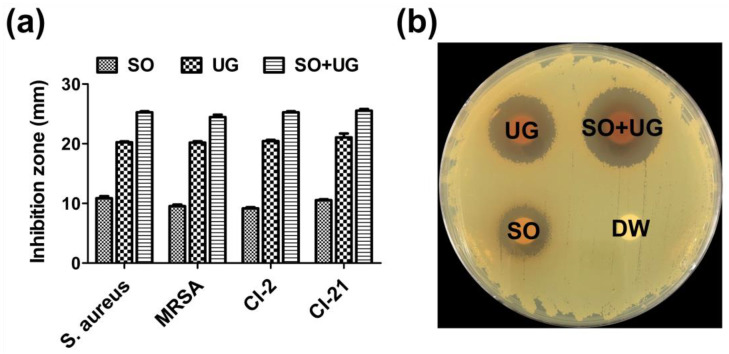
Inhibition zones of the SO + UG combination against test bacterial strains. (**a**) Comparative graph for inhibition zones; (**b**) Representative photograph (MRSA 33,591). Briefly, each bacterial suspension was adjusted to McFarland 0.5 turbidity and swabbed onto BHI plates. Then, 100 μL of each sample was loaded onto each paper disc (8 mm/diameter). The concentration of the samples was 2 mg/disc. Distilled water (DW) was served as a negative control. All data are presented as mean ± standard deviations of experiments performed in triplicate, with *p* < 0.05 indicating statistical significance.

**Figure 5 pharmaceuticals-16-01491-f005:**
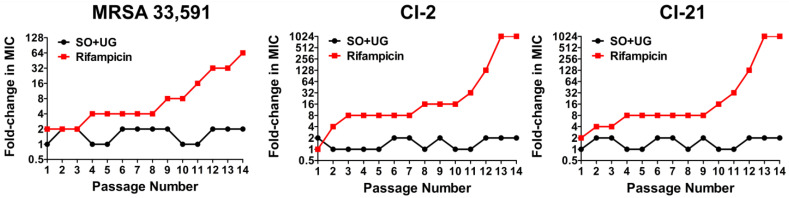
MRSA strains show no resistance to the SO + UG combination in multistep resistance selection experiments. The broth microdilution method was used to determine the MIC of the SO + UG combination and rifampicin against MRSA strains after repeated exposure. After the MIC test, the strains were taken from the sub-MIC wells and successively passaged for up to fourteen passages. The results are presented as fold-change in MIC relative to the previous passage. Resistance was defined as a more than a 4-fold increase of MIC compared to the initial MIC.

**Table 1 pharmaceuticals-16-01491-t001:** List of medicinal plants used in the present study and their pharmacological uses in traditional medicines.

Scientific Name	Common Name	Family	Parts Used	Origin	Pharmacological Uses
*Caesalpinia sappan* L.	Sappan wood	Leguminosae	Heartwood	Indonesia	Pulmonary hemorrhage and skin diseases, antibacterial, antioxidant, anti-inflammatory, hemostatic, and hepatoprotective [26,27]
*Glycyrrhiza uralensis* Fisch.	Chinese liquorice	Fabaceae	Roots	China	Respiratory and liver diseases, inflammation, antioxidant, immunoregulatory, antivirus, and antimicrobial [28]
*Sanguisorba officinalis* L.	Greater burnet	Rosaceae	Roots	China	Astringent bleeding and allergic skin diseases, anti-inflammatory, antiviral, anticancer, and antibacterial [29]
*Uncaria gambir* Roxb.	Gambir	Rubiaceae	Leaves and twigs	Indonesia	Fever and cough, bacterial/fungal infections, diabetes, inflammation, and cancer [30]

**Table 2 pharmaceuticals-16-01491-t002:** Tentative identification of the major compounds in the plant extracts.

No.	Medicinal Plants	RT (min)	[M−H]^−^ *m*/*z*	Molecular Formula	Tentative Identification
1	CS	2.98	285.07	C_16_H_14_O_5_	Brazilin
2	CS	3.10	303.08	C_16_H_16_O_6_	Protosappanin B
3	GU	4.35	549.16	C_26_H_30_O_13_	Liquiritin apioside
4	GU	9.48	821.40	C_42_H_62_O_16_	Glycyrrhizin
5	UG	3.07	289.07	C_15_H_14_O_6_	Catechin

CS: *Caesalpinia sappan* L.; GU: *Glycyrrhiza uralensis* Fisch.; UG: *Uncaria gambir* Roxb.; RT: Retention time.

**Table 3 pharmaceuticals-16-01491-t003:** Minimum inhibitory concentration (MIC) values of selected medicinal plants against test bacterial strains.

Medicinal Plants	MIC Values (μg/mL)			
	Bacterial Strains			
	*S. aureus* 29,213	MRSA 33,591	CI-2	CI-21
CS	62.5	62.5	62.5	62.5
GU	250	250	250	250
SO	250	250	250	250
UG	250	250	250	250

CS: *Caesalpinia sappan* L.; GU: *Glycyrrhiza uralensis* Fisch.; SO: *Sanguisorba officinalis* L.; UG: *Uncaria gambir* Roxb.; *S. aureus*: *Staphylococcus aureus*; MRSA: Methicillin-resistant *Staphylococcus aureus*; CI: Clinical isolate.

**Table 4 pharmaceuticals-16-01491-t004:** Fractional inhibitory concentration (FIC) values of selected medicinal plants in combination against test bacterial strains.

Bacterial Strains	Medicinal Plants		FIC Values		FIC Index	Interpretation
	A	B	FIC_A_	FIC_B_		
*S. aureus* 29,213	CS	GU	0.125	0.5	0.625	Partial synergy
	CS	SO	0.125	0.5	0.625	Partial synergy
	CS	UG	0.125	0.5	0.625	Partial synergy
	GU	SO	0.5	0.5	1	Additive
	GU	UG	0.5	0.125	0.625	Partial synergy
			0.125	0.5	0.625	Partial synergy
	SO	UG	0.25	0.25	0.5	Synergistic
MRSA 33,591	CS	GU	0.125	0.5	0.625	Partial synergy
	CS	SO	0.125	0.5	0.625	Partial synergy
	CS	UG	0.125	0.5	0.625	Partial synergy
	GU	SO	0.5	0.5	1	Additive
	GU	UG	0.5	0.125	0.625	Partial synergy
			0.125	0.5	0.625	Partial synergy
	SO	UG	0.25	0.25	0.5	Synergistic
CI-2	CS	GU	0.125	0.5	0.625	Partial synergy
	CS	SO	0.125	0.5	0.625	Partial synergy
	CS	UG	0.125	0.5	0.625	Partial synergy
	GU	SO	0.5	0.5	1	Additive
	GU	UG	0.5	0.125	0.625	Partial synergy
			0.125	0.5	0.625	Partial synergy
	SO	UG	0.25	0.25	0.5	Synergistic
CI-21	CS	GU	0.125	0.5	0.625	Partial synergy
	CS	SO	0.125	0.5	0.625	Partial synergy
	CS	UG	0.125	0.5	0.625	Partial synergy
	GU	SO	0.5	0.5	1	Additive
	GU	UG	0.5	0.125	0.625	Partial synergy
			0.125	0.5	0.625	Partial synergy
	SO	UG	0.25	0.25	0.5	Synergistic

CS: *Caesalpinia sappan* L.; GU: *Glycyrrhiza uralensis* Fisch.; SO: *Sanguisorba officinalis* L.; UG: *Uncaria gambir* Roxb.; *S. aureus*: *Staphylococcus aureus*; MRSA: Methicillin-resistant *Staphylococcus aureus*; CI: Clinical isolate; FIC index (FICI) = FIC_A_ + FIC_B_, where FIC_A_ was MIC of extract A in combination/MIC of extract A alone and FIC_B_ was MIC of extract B in combination/MIC of extract B alone. FICI are interpreted as synergistic (FICI ≤ 0.5), partial synergy (0.5 < FICI ≤ 0.75), additive (0.75 < FICI ≤ 1.0), indifferent (1.0 < FICI ≤ 4.0), or antagonistic (FICI > 4.0).

## Data Availability

Data are contained within the article and Appendix A.

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
