# Peer review of "In Vitro Synergistic Inhibitory Effects of Plant Extract Combinations on Bacterial Growth of Methicillin-Resistant *Staphylococcus aureus"

_pharmaceuticals, 2023, doi:10.3390/ph16101491_

Round 1

Reviewer 1 Report

This manuscript discusses "Synergistic Inhibitory Effects of Plant Extract Combinations on 2 Bacterial Growth of Methicillin-Resistant Staphylococcus aureus". An interesting knowledge has been reported. However, the following comments should be addressed before acceptance

comments

the novelty of the manuscript must be better emphasized

suggested to add some qualitative results in the abstract section

The importance and significance of the study should be more clearly mentioned in the introduction part

How does the concentration of the sample affect the bactericidal activity

the author should say about the molecular mechanism behind the activity

How does synergistic activity exhibit enhanced performace

How do extract preparation and dissolving agents affect the bactericidal activity

update references

After addressing all the comments, this manuscript can be acceptable for further comments

Minor revision required

Reviewer 2 Report

This is an interesting research paper examining the in vitro synergistic inhibitory effects of plant extract combinations on 2 bacterial growth of methicillin-resistant Staphylococcus aureus. The major compounds of four medicinal plants were identified by UPLC. Antibacterial bioassays were conducted to evaluate the inhibitory effects of plant extracts alone and in combination. The enhanced antibacterial activity of plant extract combinations was also studies showing that the combination SO and UG exhibited a synergistic activity. Time-kill kinetics analysis and Disc diffusion assays were conducted to determine understand the behavior of this combination.

The paper is interesting and well organized. However, the scientific soundness could be raised if the best combination tested in in vivo  

Title, should be amended to make it clear that all the experiments in this study are in vitro.

Minor editing of English language required

Reviewer 3 Report

The present manuscript entitled “Synergistic Inhibitory Effects of Plant Extract Combinations on Bacterial Growth of Methicillin-Resistant Staphylococcus aureus” by Jeong et al., describes the synergistic inhibitory effects of selected medicinal plants (Caesalpinia sappan L., Glycyrrhiza uralensis Fisch., Sanguisorba officinalis L., and Uncaria gambir Roxb.) on the bacterial growth of MRSA and its clinical isolates. The authors report an interesting work. The objective and justification of the work are clear and the supplementary data provided is too convincing for the obtained results. I congratulate the authors for their good efforts. Therefore, I recommend it for publication. However, some minor issues are detailed below which need to be addressed before its final acceptance in pharmaceuticals.

Abstract

The authors used abbreviations at first without a complete explanation.

Introduction

The authors recommended including the socio-economic status of MRSA and its impact throughout the world with proper recent reference

Methods

Why and how authors selected specifically that four medicinal plants (Caesalpinia sappan L., Glycyrrhiza uralensis Fisch., Sanguisorba officinalis L., and Uncaria gambir Roxb.), need to justify with proper reason.

Results and Discussion

Figure 1: Not able to interpret any results about the medicinal plants and better include a clear figure with more clarity

Table 2. There is no active compound from the plant Sanguisorba officinalis, then how can used this plant for combined activity study.

Table 3. Where are the antibiotic sensitivity results for CI-1 and CI-2?, without the antibiotic profiles, how come concludes CI-1 and CI-2 are MRSA strains.

The authors also discussed the importance of the MRSA in global status and its minimization of MRSA infections with future prospective.

Moderate editing of English language required.

Reviewer 4 Report

This manuscript is interesting for pharmaceutics community and could be accepted for publication. The topic is up to date and actual. Methicillin-resistant Staphylococcus aureus (MRSA) is one of the most common pathogens of healthcare-associated infections. Medicinal plants have long been used in traditional treatment of diseases or syndromes worldwide. Combined use of plant extracts could improve the effectiveness of pharmacological action by obtaining synergism, acting on multiple targets simultaneously, reducing the doses of individual components, and minimizing side effects. Authors aimed to investigate the synergistic inhibitory effects of selected medicinal plants (Caesalpinia sappan L., Glycyrrhiza uralensis Fisch., Sanguisorba officinalis L., and Uncaria gambir Roxb.) on the bacterial growth of MRSA and its clinical isolates. SO and UG extracts generated the best synergistic interaction as adjudged by checkerboard synergy assays. MICs of the individual extracts decreased by 4-fold, respectively. SO+UG combination was further evaluated for its effects on bacterial growth inhibition, MBC/MIC ratio, and time-kill kinetics. The results indicate that SO+UG combination synergistically inhibited the bacterial growth of MRSA strains with bactericidal effects. SO+UG combination also exhibited more potent effects against clinical isolates. In multi-step resistance selection experiments, both standard and isolates of MRSA showed no resistance to SO+UG combination. Author’s data suggest that using plant extract combinations could be a potential strategy to treat MRSA infections. The subject addressed in this article is worthy of investigation. The information presented is new. The methodology of research is appropriate. The conclusions supported by the data. The manuscript is good illustrated and interesting to read. I have only one suggestion for minor revision: some more detailed perspectives about the future research could be formulated in conclusions.

Reviewer 5 Report

In the manuscript "Synergistic Inhibitory Effects of Plant Extract Combinations on Bacterial Growth of Methicillin-Resistant Staphylococcus aureus" the synergistic effect of several plant extracts on MRSA is described. Although the topic is very interesting, the article is not suitable for publication in this form because there are shortcomings that need to be explained.

First of all, since we are talking about plant extracts, I am wondering if the extracts are colored. Namely, it greatly hinders the performance of spectrophotometry, so I'm interested in which blank samples you used.

The growth curve in Figure 3b shows that the lag phase lasts 8-10 hours, which is impossible for this bacterium. The result does not match with Figure 3b where CFU indicates that the increase in the number of bacteria occurs much faster. And it must be faster because the generation time of staphylococci is not that long.

If the reading was disturbed due to the color of the extract, then the results of the synergy are also questionable. As for the checkerboard assay, a more reliable method of determining inhibition is resazurin or plating on agar.

Can you explain multistep resistance selection in more detail?

Round 2

Reviewer 2 Report

The authors improved the manuscript

The new version can be considered for publication 

Reviewer 5 Report

The authors explained all the shortcomings. I have no further comments. For another time, first do the growth curve without treatment and then start with treatments. I have no further comments.